# SARS-CoV-2 Did Not Spread Through Dental Clinics During the COVID-19 Pandemic in Japan

**DOI:** 10.3390/idr17030070

**Published:** 2025-06-13

**Authors:** Yasuhiro Tsubura, Yuske Komiyama, Saori Ohtani, Toshiki Hyodo, Ryo Shiraishi, Shuma Yagisawa, Erika Yaguchi, Maki Tsubura-Okubo, Hajime Houzumi, Masato Nemoto, Jin Kikuchi, Chonji Fukumoto, Sayaka Izumi, Takahiro Wakui, Koji Wake, Hitoshi Kawamata

**Affiliations:** 1Department of Oral and Maxillofacial Surgery, School of Medicine, University of Dokkyo Medical, 880 Kitakobayashi Mibu Shimotsuga, Tochigi 321-0293, Japan; y-tsubura@dokkyomed.ac.jp (Y.T.); y-komi@dokkyomed.ac.jp (Y.K.); ohtani79@dokkyomed.ac.jp (S.O.); hyodo14@dokkyomed.ac.jp (T.H.); ryo-s@dokkyomed.ac.jp (R.S.); s-yagi@dokkyomed.ac.jp (S.Y.); eri-yagu@dokkyomed.ac.jp (E.Y.); makio@dokkyomed.ac.jp (M.T.-O.); chonji-f@dokkyomed.ac.jp (C.F.); saya@dokkyomed.ac.jp (S.I.); 2Yasu Kazu Charm Dental Clinic, 2-4-35 Takinohara Utsunomiya, Tochigi 320-0846, Japan; 3Department of Emergency and Critical Care Medicine, School of Medicine, University of Dokkyo Medical, 880 Kitakobayashi Mibu Shimotsuga, Tochigi 321-0293, Japan; houzumi@dokkyomed.ac.jp (H.H.); m-nemoto@dokkyomed.ac.jp (M.N.); k-jin@dokkyomed.ac.jp (J.K.); wake@dokkyomed.ac.jp (K.W.)

**Keywords:** SARS-CoV-2, COVID-19, aerosol, dentistry, oral management

## Abstract

Background: Dental professionals were thought to have the most significant risk of coronavirus infection during the pandemic. Since the first Coronavirus Disease 2019 (COVID-19) patient was detected in Japan in January 2020, Japan has faced several waves of Severe Acute Respiratory Syndrome Coronavirus 2 (SARS-CoV-2) infections. However, no cluster of SARS-CoV-2 infections associated with dental procedures has been reported in Japan. In this study, we aimed to investigate the actual status of SARS-CoV-2 infection during the pandemic through antibody testing for dental professionals. We further investigated saliva and oral management-related aerosol to estimate the risk of virus transmission during dental procedures. Methods: SARS-CoV-2 antibody titer in the blood of dental professionals and their families was determined during the pre-vaccinated period of the SARS-CoV-2 wave to see the history of infection in Japan. Viral loads in saliva and in the aerosol generated during the oral management of COVID-19 patients were detected by RT-qPCR. Results: The antibody testing of dental healthcare providers during the early phases of the pandemic in Japan revealed low antibody positivity, which supported the low incidence of infection clusters among dental clinics. The aerosol generated during dental procedures may contain trace levels of SARS-CoV-2, indicating the risk of transmission through dental procedures is limited. Therefore, SARS-CoV-2 did not spread through dental clinics. Conclusions: Very few SARS-CoV-2 infections were observed in dental professionals who took appropriate infection control measures in the early period of the pandemic. Performing dental procedures using standard precautions seems to be sufficient to prevent SARS-CoV-2 infections.

## 1. Introduction

In a 15 March 2020 article titled “The Workers Who Face the Greatest Coronavirus Risk”, *The New York Times* listed dental hygienists, dental assistants, dentists, and other dental professionals among the professions with the most significant risk of coronavirus infection [1]. The first COVID-19 patient was detected in Japan in January 2020. When the first declaration of a state-of-emergency order was released, many dental professionals in Japan closed or limited their practices as SARS-CoV-2 infections began surging from April to June 2020 in Japan. However, since no significant clusters were reported at dental clinics in Japan, most clinics restarted normal operations when the declaration was lifted. The Japanese government subsequently issued a state of emergency three times (from 8 January to 21 March 2021; from 25 April to 20 June 2021; and from 12 July to 30 September 2021), but did not implement a lockdown with legal enforcement. Additionally, starting around April 2020, the Japanese government issued guidelines to all citizens, including requests to refrain from non-essential outings, wear masks when going out, and avoid enclosed spaces, crowded places, and situations involving close contact. The Alpha variant of SARS-CoV-2 was first detected in December 2020 in Japan. The Delta variant of SARS-CoV-2 was first detected in Japan toward the end of March 2021, and the spread into the community began in April 2021. Since then, Japan has faced several waves of SARS-CoV-2 infections, plus an eighth wave caused by the Omicron variant in May 2023. In May 2023, SARS-CoV-2 infections were downgraded to an infectious disease that does not require notification in Japan, and although the exact number of patients since then is unknown, several waves of infections have repeatedly occurred in Japan.

Statistics on clusters occurring through the fifth wave in Japan studied by the Japanese Ministry of Health, Labor and Welfare indicate that, as of July 2021, 1225 clusters involving medical institutions and 1680 clusters involving care facilities for elderly persons or children with disabilities had occurred [2]. Although sporadic infections occurred in dental professionals at dental clinics during this period, no cluster of SARS-CoV-2 infections associated with dental procedures had been reported in Japan until now [3]. Regarding whether dental professionals are at higher risk of lingering SARS-CoV-2 infection, dental professionals have continued to provide care and treatment for infected individuals for three years in Japan. Fortunately, the initial predictions by *The New York Times*, however, have failed to happen. Dental professionals appear not to have caused any infections or clusters through horizontal transmission between clinic patients. SARS-CoV-2 is present in large amounts in an infected person’s saliva [4], and dental procedures are aerosol-producing procedures that involve saliva at various levels [5]. In the current consensus, SARS-CoV-2 is reported to transmit through direct contact and droplets, and possibly droplet nuclei (airborne) in some conditions [6]. When considering the nature and transmission of SARS-CoV-2, we were led to question why dental professionals, who rely primarily on standard precautions for contact infection, have remained relatively unscathed and why clusters have not been seen at dental clinics.

Puzzled by these questions, we performed antibody testing to determine the level of previous infections in dental professionals during the pre-vaccine period of SARS-CoV-2, corresponding to the time from the first to third waves in Japan. The virus load in aerosols produced during dental procedures performed on COVID-19 patients and in their saliva was also determined to answer why SARS-CoV-2 did not spread through dental clinics.

## 2. Materials and Methods

### 2.1. Determination of IgG/IgM Antibody Titer Against SARS-CoV-2 in the Blood (Evidence for Previous Infections) in Dental Professionals and Their Families During the SARS-CoV-2 Wave

This portion of the study was conducted with the approval of the Clinical Research Management Center, Dokkyo Medical University (Approval No. R-35-21J, 21 June 2020). Informed consent was obtained from all participants following the “Ethical Guidelines for Medical and Biological Research Involving Human Subjects” in Japan.

The participants were oral and maxillofacial surgeons and dental hygienists practicing in the Department of Oral and Maxillofacial Surgery of Dokkyo Medical University Hospital. Furthermore, their families, as well as associated dentists, dental hygienists, nurses, dental assistants, dental technicians, and administrative staff from their clinics, were examined. We obtained informed consent for participating in this study from the participants, and blood samples were immediately collected and tested. Subsequently, we obtained informed consent from their families and the dentists, dental hygienists, nurses, dental assistants, dental technicians, and administrative staff practicing in the associated clinics, and blood samples were collected and tested immediately.

The antibody testing was performed using the GenBody COVID-19 IgM/IgG Test^®^ rapid antibody detection kit for SARS-CoV-2 (GenBody Inc., Chungcheongnam-do, Korea). A 20 µL amount of whole blood collected from each participant was applied to the test plate. Antibody status was assessed after 10 to 20 min. The kit’s sensitivity, specificity, and accuracy were as follows: sensitivity 50% on Day 1–6, 91.7% after Day 7; specificity 97.5%; and accuracy 95.2% on Day 1–6, 96.5% after Day 7.

SARS-CoV-2 antibody tests were performed three times on pre-vaccinated dental professionals to coincide with periods of high SARS-CoV-2 prevalence in Japan (as determined according to the number of PCR-positive people in public data on the website of the Japanese Ministry of Health, Labor, and Welfare). The first round of tests was performed from 27 May to 3 July 2020, the second from 23 September to 30 October 2020, and the third from 31 January to 3 April 2021 (Figure 1). The antibody positivity rate in dental professionals and their families was calculated.

The Alpha variant was first detected December 2020 in Japan. The Delta variant was first detected in Japan toward the end of March 2021, and spreading into the community of the Delta variant began in April 2021.

### 2.2. Sample Collection of Saliva and Aerosol During Oral Management for Hospitalized COVID-19 Patients

We performed this part of the study with the approval of the Clinical Ethics Committee of Dokkyo Medical University Hospital (Approval No. R-45-2 14 April 2021). The participants were COVID-19 patients hospitalized for treatment who were referred for oral management by doctors in the Department of Emergency and Critical Care Medicine. All participants had moderate-to-severe respiratory symptoms and required respiratory management. Participants capable of consenting were informed orally about the study and then asked to consent. When a participant could not give consent, a family member was asked to consent on their behalf.

We conducted an oral examination before the management. Oral management included oral cleaning, masticatory and swallowing rehabilitation to maintain and improve oral functions, education for patients, family members, and healthcare professionals. Then, No. 51A filter paper (Advantec Toyo Kaisha, Ltd., Tokyo, Japan), cut to 15 × 15 mm and autoclaved, was placed in the buccal vestibule or floor of the mouth for 1 min. The wetted filter paper was collected in a 1.5 mL tube. For the aerosol sample collection, the filter paper and a gelatin filter (Sartorius Corporate Administration GmbH, Gottingen, Germany) were attached to the center of the inlet of extraoral suction equipment before the start of the oral management. The vacuum inlet was placed approximately 20 cm from the patient’s mouth. Aerosol suction was maintained during the examination and the management. Then, dental plaque and detached epithelium were removed by tooth brushing and wiping, and dental scaling was performed with a VIVAase ultrasonic scaler (Nakanishi Inc., Kanuma, Japan). After dental scaling, residual debris was removed with another round of brushing and wiping. Finally, a moistening gel (Pepti-Sal Gentle MouthGel, T & K Co., Tokyo, Japan; Hinora, Otsuka Pharmaceutical Factory, Tokushima, Japan) was applied to the oral mucosa to moisten the mouth and lips. The oral management procedure took about 15 min per patient. After the management, the gelatin filter and filter paper were collected as aerosol samples. The saliva and aerosol samples were soaked in 250 µL of normal saline and stored at −25 °C until analysis.

### 2.3. Detection of SARS-CoV-2 Viral Genome with RT-qPCR

Viral genome detection was conducted according to the instructions for using the SARS-CoV-2 Direct Detection RT-qPCR Kit (Takara Bio Inc., Shiga, Japan). Briefly, 8 µL of each sample was mixed with Solution A. The mixture was heat-treated and then analyzed with RT-qPCR. SARS-CoV-2 RNA US N1/N2 of a known concentration was used as the positive control (Takara Bio Inc.). RNase-free water was used as the negative control. The primers used with the kit were detection primers listed in “2019-Novel Coronavirus (2019-nCoV) Real-time RT-PCR Panel Primers and Probes” of the United States Centers for Disease Control and Prevention [7].

### 2.4. Statistical Analysis

Descriptive statistics were determined for the participants’ demographics in the antibody and PCR studies. A chi-squared test with a significance level of 0.05 was used to statistically test the data of the antibody study of the participants. Student’s *t*-test with a significance of 0.05 was used to compare Ct values associated with virus detected in saliva and aerosol samples. In the scatter plot of viral loads, values below the detection limit on Ct value analysis were indicated as 0, and values below the detection limit in copy number analysis were indicated as 0.01.

## 3. Results

### 3.1. Antibody-Positive Rate for SARS-CoV-2 in Dental Professionals

All participants’ ages were over 20 at the time of consent (Table 1). Those who refused or were found unsuited to participate by the principal investigator (HK) were excluded from the study. SARS-CoV-2 prevalence from the first to third waves in Japan is shown together with the timing of the antibody tests in Figure 1. Japanese people used only social and personal infection prevention measures through the third wave. Vaccines for SARS-CoV-2 became available just before the fourth wave hit in Japan. Antibody tests in this study were performed at the end of the first, second, and third waves when antibodies from previous infections would have been detectable if infected with the virus. Worthy of note is that the antibody tests were performed before vaccination in Japan; therefore, they did not indicate rises in antibody titer from vaccination. We show the results of the antibody tests in Table 2, which were performed to coincide with times of prevalence. IgG antibodies were detected in 2 of the 424 tests performed. One affected participant was a dentist, and the other test was from a cohabiting family. None of the samples were positive for IgM antibodies. Since only two participants tested positive for antibodies, comparing the frequencies of previous infections among occupations, place of residence, or according to what infection prevention measures were in place at different clinics was impossible. The results, however, show that very few dental professionals had a history of infection before the fourth wave in Japan.

### 3.2. Viral Loads in the Saliva or Oral Management-Related Aerosol of COVID-19 Patients

We analyzed the viral loads in samples of saliva or oral management-related aerosols collected from 24 COVID-19 patients (Table 3). The mean age was 59.0 (42–89) years. Twenty-two of the participants were men, and two were women. Samples were collected during the 39 oral management sessions performed. The SARS-CoV-2 detected at the time of the management in these patients was Variant 20I (Alpha, V1)/B.1.1.7 in 12 patients and Variant 21A (Delta)/B.1.617.2 in 11 patients. The variant type of one patient was not determinable. Only three of the patients had no medical history. Eighteen of the twenty-four patients (75%) had a history of hypertension, diabetes mellitus, cardiovascular disease, obesity, dyslipidemia, or respiratory disease. Most of the patients had comorbidities. The days until sample collection from the positive result ranged from 3 to 46 days for SARS-CoV-2 and was, on average, 14.6 days. A total of 22 of the 24 patients required mechanical ventilation (noninvasive positive pressure ventilation: 4 patients, intubation: 18 patients [intraoral intubation: 10 patients, tracheostomy: 8 patients]). Two patients required extracorporeal membrane oxygenation along with invasive mechanical ventilation. Most patients had ventilation-associated oral dryness and biofilms consisting of dental plaque, tartar, and detached epithelium. Oral management by dental scaling and tooth brushing was possible for 17 of the 24 patients (70.8%). Management was possible only through tooth brushing for 6 of the 24 patients (25%) (Table 3). Viral loads in the saliva and aerosol samples are shown regarding RT-qPCR Ct values and copy numbers (Figure 2). Mean Ct values were 34.76 [95% confidence interval (CI): 32.45, 37.07] for the saliva samples and 38.99 [95% CI: 37.28, 41.28] for the aerosol samples. Copy numbers were estimated to be 969,495 [95% CI: −519,755, 2,458,745] per 250 µL for the saliva samples and 1656 [95% CI: −102, 3413] for the aerosol samples. Although the viral loads in the saliva and aerosol samples are not amenable to comparison, mean Ct values in the aerosol were near the limit of detection, with copy numbers on the order of 10^6^ to 10^7^ in the saliva samples but 10^3^ in the aerosol samples, which represents a 1000- to 10,000-fold decrease.

## 4. Discussion

Our antibody tests performed on dental professionals showed a low positivity rate in the period just after SARS-CoV-2 infection peaked in the general population. Moreover, in Japan, no clusters associated with a dental procedure in a dental setting have been reported. In 2020, to estimate the spread of SARS-CoV-2 in Japan, some companies and agencies conducted an epidemiological survey of infection by antibody testing. The test conducted by the Ministry of Health, Labor, and Welfare to evaluate the performance of antibody test kits showed an antibody positivity rate of 0.1% to 1.35% in Tokyo, where infections were prevalent at the time [8,9]. Large-scale antibody testing conducted by SoftBank Group Corp (SoftBank Group Corp, Minato Ward, Japan). showed an antibody positivity rate of 0.43% in the overall sample, 1.79% in healthcare professionals, 0.89% in dental assistants, and 0.75% in dentists [10]. Note that the antibody positivity rate among close contacts is thought to be 11.76%. The low antibody positivity (i.e., previous infection) rate of 0.47% identified in dental professionals through the third wave follows the rates seen in nationwide, large-scale testing programs, reflecting actual levels. Santigli et al. recently reported that the seroprevalence of SARS-CoV-2 IgG antibodies among dental healthcare workers (dentists, dental assistants, reception staff, etc.) was reported to be 13.49% (95% confidence interval: 9.15–18.52%) overall [11]. This systematic review and meta-analysis analyzed data from 6083 dental healthcare workers across 10 observational studies in seven European countries and Brazil from 2020 to 2021, with no data from Japan included. These results indicate a higher incidence compared to other healthcare workers (7–9%) and the general population (9.5%), suggesting that dental healthcare workers may be at higher risk of infection. These findings suggest that the antibody positivity rate among dental healthcare workers may vary depending on factors such as country, region, the status of the pandemic, and the rigor of infection control measures. The strengths of our study include the fact that the study could have been conducted during the pandemic period before the vaccination in Japan had certain impacts.

Several issues face the provision of dental procedures during the SARS-CoV-2 pandemic: cells of the lingual mucosa and salivary glands harbor the virus, meaning that saliva is a potential source of infection [4]; dental professionals are at risk of exposure to virus-laden aerosols when treating patients; the diffusion of aerosols in the air in clinics poses the risk of horizontal transmission between patients; difficulty to screen for asymptomatic/subclinical infections (which nonetheless feature viral shedding), meaning there is no way to determine the risk of infection before a dental procedure is administered. These issues were the source of concern for the spread of viral infections via dental procedures. However, no clusters associated with a dental procedure in a dental setting have occurred in Japan. Why the aerosols produced during dental procedures did not spread infection remains unknown. Microbial infections have been conventionally classified and discussed as being transmitted via direct contact, droplets, and droplet nuclei; the concept of aerosol-transmitted infection was proposed for SARS-CoV-2 infections, which have occurred in a manner not attributable to direct contact or droplets [6]. An aerosol is a mixture of tiny solid or liquid particles (of unspecified sizes) floating in the air. However, aerosols in the context of infections are generally defined as micro-droplets measuring no more than 5 µm in diameter. The droplets and aerosol generated from ultrasonic dental scalers may disperse upwards of 6 feet from the source [12], and aerosols may vaporize as they travel through the air at average temperatures and humidity levels. The part remaining following evaporation corresponds to droplet nuclei as conventionally defined, but the trace quantities of virus remaining in desiccated droplet nuclei might be insufficient to cause infections for SARS-CoV-2 infections [13].

The present study showed that relatively high viral loads were detectable in the saliva of patients with severe COVID-19 even several days after they were found positive for the virus. Although it is known that SARS-CoV-2 is present in the saliva of infected people, it is unclear whether the virus is present when saliva is secreted from infected salivary gland cells [14], the virus contaminates saliva from infected upper airway mucosa, or the virus contaminates saliva from lower airway secretions. In the present study, aerosols generated over 15 min were trapped in a piece of filter paper 20 cm from the patient’s mouth. The virus was detectable in only 10 of the 39 samples collected, at relatively low levels. Usually, symptomatic COVID-19 patients do not visit a local dental clinic. Most patients visiting dental clinics are expected to be asymptomatic or presymptomatic with COVID-19. Asymptomatic or presymptomatic COVID-19 patients are now known to shed almost equal amounts of the virus from symptomatic patients [15]. We showed that even when a substantial amount of the virus is present in the saliva, only a small amount remains in the aerosol produced in dental settings. Meethil et al. also reported that aerosol from dental settings carries less viral load than saliva [16]. A report by Katsumi et al. from the Sendai Municipal Institute of Public Health investigated the relationship between viral isolation and PCR detection of nasal swabs collected from symptomatic patients. It estimated that one infective particle could be present in 100 to 1000 viral copies [17]. Based on their reports combined with our current findings, it is suggested that no more than ten infective particles are present in such aerosol. Therefore, the aerosol generated during dental procedures on COVID-19 patients is unlikely to cause infection. This could be part of the reason why dental clinics have not spread SARS-CoV-2 infection during the pandemic.

We believe that we can prevent infecting others through established measures, even if the aerosol contains SARS-Co-2 from the saliva of asymptomatic or presymptomatic COVID-19 patients during dental procedures. Especially in dental settings, appropriate personal protective equipment and intra- and extraoral suction equipment are beneficial. With an appreciation of these findings, oral surgeons and other dental professionals can maintain their patients’ oral health with the proper preventative measures, even for future pandemics brought on by microorganisms which show direct contact or droplet infections. From a public health standpoint, it is important to accurately assess the situation in a pandemic, and countermeasures must be taken accordingly. However, in this SARS-CoV-2 pandemic, when the mode of transmission, infectivity, and methods of infection prevention were not clear, over-prevention measures had to be taken. It is important to make effective use of the data we have obtained in this study for future pandemics, and at the same time, it is important to establish a system to collect and analyze data on infection status and infection patterns quickly and appropriately for all healthcare workers, including dental professionals.

## 5. Limitation

This study is an observational study and has the following limitations. Antibody testing was conducted on dental professionals and their associates, and no other subjects were examined as controls. The rapid antibody test had a high specificity (97.5%) but a sensitivity of 50% on Days 1–6 after infection and 91.7% on Day 7 after infection. Furthermore, the number of individuals who underwent all three antibody tests was limited. Aerosol samples were collected during actual clinical procedures, oral management, and at various time points after SARS-CoV-2 infection, and environmental factors were not consistent. The ability of extraoral vacuums to collect aerosols generated during dental treatment, the detection threshold and detection limit of microorganisms in aerosols, and the efficiency of supplementation of microbial particles, such as viruses, were not studied in advance of this study. To the best of our knowledge, there are no established standards for detecting microorganisms in aerosols. These might also be potential limitations of this study.

## Figures and Tables

**Figure 1 idr-17-00070-f001:**
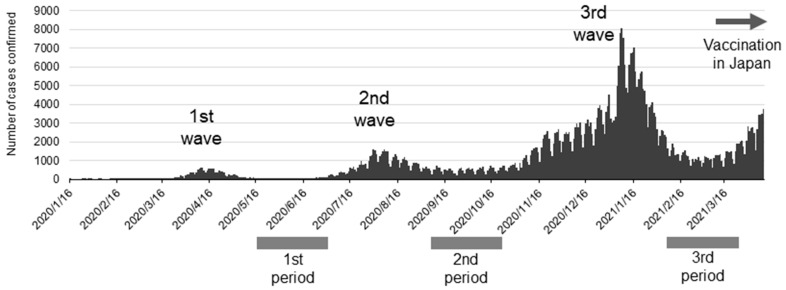
Time course of new SARS-CoV-2 infections in Japan and the timing of antibody tests in the study.

**Figure 2 idr-17-00070-f002:**
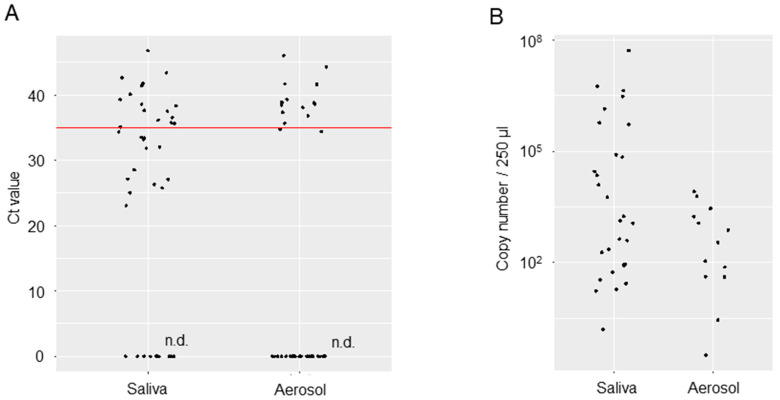
Viral loads in the saliva or aerosol samples of COVID-19 patients. (**A**) Ct values of saliva and aerosol samples. Aerosol samples were captured on a 2.25 cm^2^ piece of filter paper. The red line represents the cut-off value for social quarantine measures (a Ct value of 35 was considered in Japan). (**B**) Estimated copy numbers in the total sample volume (250 µL) are shown.

**Table 1 idr-17-00070-t001:** Examinees’ demographic characteristics *.

Characteristic	
Age—y (range) †	39.8 (20–77)
Age—y, no./total no. (%)	
20 to 29	74/304 (24.3)
30 to 39	83/304 (27.3)
40 to 49	76/304 (25.0)
50 to 59	48/304 (15.8)
60 to 69	18/304 (5.9)
70 to 79	4/304 (1.3)
Not specified	1/304 (0.3)
Sex—no./total no. (%)	
Male	80/304 (26.3)
Female	224/304 (73.7)
Occupation—no./total no. (%)	
Dental hygienist	116/304 (38.2)
Dentist/oral surgeon	97/304 (31.9)
Dental assistant	40/304 (13.2)
Medical clerk	28/304 (9.2)
Family member	9/304 (3.0)
Dental technician	8/304 (2.6)
Nurse	5/304 (1.6)
Others	1/304 (0.3)
Location of healthcare facility—no./total no. (%)	
Tochigi	162/304 (53.3)
Gunma	42/304 (13.8)
Saitama	34/304 (11.2)
Tokyo	34/304 (11.2)
Kanagawa	22/304 (7.2)
Fukushima	9/304 (3.0)
Ibaraki	1/304 (0.3)
Chiba	1/304 (0.3)
Others	6/304 (2.0)

* Percentages may not total 100 because of rounding. † Data were missing for one examinee.

**Table 2 idr-17-00070-t002:** Determination of the antibody against SARS-CoV-2.

	1st Period	2nd Period	3rd Period	Total
All participants	2/291 (0.69%)	0/31 (0%)	0/102 (0%)	2/424 (0.47%)
Dentist	1/92 (1.09%)	0/30 (0%)	0/46 (0%)	1/168 (0.60%)
Dental hygienist	0/112 (0%)	0/0 (0%)	0/40 (0%)	0/152 (0%)
Dental assistant	0/40 (0%)	0/0 (0%)	0/5 (0%)	0/45 (0%)
Medical clerk	0/25 (0%)	0/0 (0%)	0/8 (0%)	0/33 (0%)
Family	1/9 (11.11%)	0/1 (0%)	0/1 (0%)	1/11 (9.09%)
Dental technician	0/8 (0%)	0/0 (0%)	0/1 (0%)	0/9 (0%)
Nurse	0/5 (0%)	0/0 (0%)	0/0 (0%)	0/5 (0%)
Others	0/0 (0%)	0/0 (0%)	0/1 (0%)	0/1 (0%)

**Table 3 idr-17-00070-t003:** Patients’ demographic characteristics at the time of oral healthcare.

Patients Number	Age (y)	Sex	SARS-CoV-2 Strain	Days of Sampling from Positive Confirmation	Type of Respiratory Management	Method of Oral Management	Ct Values (Saliva)	Ct Values (Aerosol)
1	66	M	alpha	5	IPPV (intraoral intubation)	Brushing, Scaling	25.8	35.67
2	66	M	alpha	4	IPPV (tracheotomy)	Brushing, Scaling	37.665	-
3	64	M	alpha	12	HFOT	Brushing, Scaling	41.41	-
4	53	M	alpha	13	IPPV (tracheotomy)	Brushing, Scaling	41.765	-
5	59	M	alpha	16	NIPPV	Brushing, Scaling	-	37.35
6	53	M	alpha	7	IPPV (intraoral intubation) with ECMO	Brushing, Scaling	26.33	-
7	72	M	alpha	8	NIPPV	Brushing, Scaling	-	-
8	53	M	alpha	5	IPPV (tracheotomy)	Brushing, Scaling	31.89	-
9	73	M	alpha	7	IPPV (intraoral intubation)	Brushing, Scaling	35.62	-
10	54	M	alpha	2	IPPV (intraoral intubation)	Brushing, Scaling	39.32	-
11	53	M	alpha	9	IPPV (tracheotomy)	Brushing, Scaling	-	-
12	53	M	alpha	5	IPPV (tracheotomy)	Brushing, Scaling	-	-
13	47	M	delta	4	IPPV (intraoral intubation)	Brushing, Scaling	-	-
14	52	M	alpha	16	NIPPV	Brushing, Scaling	36.125	38.91
15	48	F	alpha	47	IPPV (intraoral intubation)	Brushing, Scaling	36.555	-
16	50	M	delta	5	HFOT	Brushing	46.79	-
17	42	M	delta	4	IPPV (intraoral intubation) with ECMO	Brushing	40.135	34.415
18	42	M	delta	12	IPPV (intraoral intubation) with ECMO	Brushing	25.05	-
19	46	M	alpha	13	IPPV (intraoral intubation)	Brushing	35.755	46.05
20	52	M	delta	16	IPPV (intraoral intubation)	Brushing	33.4	-
21	89	M	delta	7	IPPV (tracheotomy)	Brushing, Scaling	27.1	-
22	62	M	alpha	8	IPPV (tracheotomy)	Brushing	32.01	-
23	42	M	delta	5	IPPV (tracheotomy)	Brushing	35.095	38.435
24	76	M	delta	7	IPPV (intraoral intubation)	Brushing	23.095	-
25	48	M	delta	2	IPPV (tracheotomy)	Brushing, Scaling	-	-
26	42	M	delta	9	IPPV (tracheotomy)	Brushing, Scaling	-	-
27	62	M	alpha	5	IPPV (tracheotomy)	Brushing, Scaling	38.35	-
28	62	F	-	4	IPPV (intraoral intubation)	Brushing	33.14	-
29	76	M	delta	16	IPPV (intraoral intubation)	Brushing, Scaling	28.56	-
30	57	M	delta	47	IPPV (intraoral intubation)	Brushing	34.4	-
31	76	M	delta	5	IPPV (intraoral intubation)	Brushing, Scaling	33.5	38.085
32	57	M	delta	4	IPPV (tracheotomy)		43.425	36.805
33	62	M	alpha	12	IPPV (tracheotomy)		-	41.645
34	47	M	delta	13	NIPPV	Brushing, Scaling	42.62	44.295
35	47	M	delta	16	NIPPV	Brushing, Scaling	-	-
36	62	M	alpha	7	IPPV (tracheotomy)	Brushing, Scaling	-	-
37	57	M	delta	8	IPPV (tracheotomy)	Brushing, Scaling	27.12	38.805
38	57	M	delta	5	IPPV (tracheotomy)	Brushing, Scaling	37.49	41.61
39	58	M	delta	7	HFOT	Brushing, Scaling	38.595	34.762

IPPV; invasive positive pressure ventilation, HFOT; high-flow oxygen therapy, NIPPV; noninvasive positive pressure ventilation, ECMO; extracorporeal membrane oxygenation.

## Data Availability

All data are included in the manuscript.

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
