# Peer review of "SARS-CoV-2 Did Not Spread Through Dental Clinics During the COVID-19 Pandemic in Japan"

_2036-7449, 2025, doi:10.3390/idr17030070_

Round 1

Reviewer 1 Report

Comments and Suggestions for Authors
  1. In the introduction, add the individual and collective preventive measures implemented by the government in response to the pandemic.
  2. It is recommended that Table 1 be included in the results section. Also, there are 304 participants; in this regard, please indicate the reason for changing the denominator to "304" or "307."
  3. Lines 118 to 121 would be appreciated if this information could be included; however, it is suggested that it be included in the introduction.
  4. It is noteworthy that nine family members were included in the blood sample collection. Please indicate if the remaining family members were not included or what happened. Likewise, in the family unit, no staff member has a pediatric population. Please specify.
  5. It is recommended that you detail the method of recruitment and selection of participants, as well as the time of sample collection. As they belong to a University Department, staff either came to your center to collect the samples or were summoned to a specific area for this purpose. Please specify this time point for requesting informed consent and collecting samples.
  6. Although Table 2 provides the details for each patient, two options are recommended: 1) include this table as supplementary material, and 2) create a summary table with the descriptive analysis for these 39 individuals. Also, include it in the Results section.
  7. In Table 3, please indicate the difference in the total (291 participants) compared to the total in Table 1 (307). Please also indicate the reasons why not all participants were sampled in the second and third samples.
  8. Recommended discussion structure: First paragraph: Essential interpretation of the main result with scientific support. Second paragraph: Compare and contrast in light of other studies; formulate hypotheses about the results, highlight unexpected results and formulate hypotheses, and indicate the strengths and limitations of the study. Third paragraph: Summarize the hypothesis and purpose of the study, your contributions to the study, and the most relevant result. Indicate unanswered questions that lead to other types of studies. In this regard, we are keen to include points such as unexpected results, strengths and limitations of the study, and the conclusion.
  9. In my personal opinion, this type of study has been documented in the general population or in healthcare workers (physicians and/or nurses). However, in the field of dentistry, I have not had the opportunity to read a similar manuscript. In this regard, in the discussion, reference 10 is mentioned regarding a study to determine antibodies. Are there more studies in this population? Not only in Japan, but elsewhere, to be able to quantify the results with the containment measures implemented.

Reviewer 2 Report

Comments and Suggestions for Authors

This manuscript addresses an interesting topic, the real-world transmission risk of SARS-CoV-2 in dental clinics. By combining seroprevalence data from dental professionals with direct aerosol and saliva viral load measurements in hospitalized COVID-19 patients, the authors make a compelling case that transmission in dental settings was minimal under standard precautions. The topic is relevant, the data are robust, and the findings support current infection prevention policies.

Please check the following comments:

  1. The title may benefit from a little upgrade. For example including the mention of COVID-19: "SARS-CoV-2 Did Not Spread Through Dental Clinics During the COVID-19 Pandemic in Japan"
  2. Although the introduction is adequate, a more structured hypothesis could be included. something like "We hypothesized that adherence to standard infection control practices prevented SARS-CoV-2 transmission in dental clinics."
  3. I suggest switching some stuff for better readability. Specifically, I would move table 1 and figure 1 in the results section under the subtitle demographics or overview. Table 2 could also go to this subtitle or it could be put in a supplementary file. This way the materials section is more condensed and unified, with tables and figures, which I understand come directly from your study, may be grouped better in the results section.
  4. Clarify use of “oral management” early in the methods, it might be unfamiliar to an international audience.
  5. Add a limitations section mentioning the nature of the study (retrospective, observational), the low sensitivity of rapid tests, especially in the earlier period, lack of control group, variability of aerosol sample collection timings, how was missing data handled.
  6. The manuscript does not detail the detection threshold or limit of detection in aerosols, the efficiency of viral particle capture or validation/ control, an objective method of assessment regarding compliance to control measures. Any missing information should be at least acknowledged as a potential limitation.
  7. In the discussion section I think another important topic would be the impact and involvement of variants on aerosol viral load.
  8. At the end of the discussion section, before limitations, a paragraph discussing further perspective and implications could be added. this paragraph could contain implications on public health, future preparedness,
  9. English corrections: 

A.  "at pandemic period" -> during the pandemic. (check throughout the document)

B. healthcare is generally written in a single word not separate

C. repetitive phrasing of “pandemic period”, try some alternatives

D. “SARS-CoV-2 is reported to transmit through contagious and droplet, and possibly droplet nuclei...” is awkward

E. staffs does not have a plural (check throughout the document)

F. some typos here and there: "cilnic"
